# A Comparison of Paddle Forces between Whitewater and Flatwater Training in C1 Canoe Slalom

**DOI:** 10.3390/jfmk9030167

**Published:** 2024-09-17

**Authors:** James M. Wakeling, Stanislava Smiešková, Matej Vajda, Jan Busta

**Affiliations:** 1Department of Biomedical Physiology and Kinesiology, Simon Fraser University, Burnaby, BC V5A 1S6, Canada; 2Hamar Institute for Human Performance, Faculty of Physical Education and Sports, Comenius University, 814 99 Bratislava, Slovakia; matej.vajda@gmail.com; 3Faculty of Physical Education and Sport, Charles University, 110 00 Prague, Czech Republic; buster@centrum.cz

**Keywords:** canoe slalom, force, paddle, stroke, training

## Abstract

Background/Objectives: Becoming an elite canoe slalom athlete requires thousands of hours of training, spread over many years. It is difficult to assess the correct balance between flatwater and whitewater training because differences in the paddle forces on these terrains are not known. The aim of this study was to describe paddle forces during canoe slalom training on flatwater and whitewater courses for the C1 canoe category. Methods: Paddle forces for twenty C1 canoe slalom athletes were quantified during all-out figure-of-eight tests on a flatwater course and during race simulations on a whitewater course. Paddle forces were measured using strain gauges embedded in the paddle shaft and quantified by their force, impulse, and stroke durations. Results: The mean force during the pull phase of the paddle strokes was not significantly different between the flatwater and whitewater courses; however, the longer pull phase durations led to a greater pull phase impulse when paddling on the whitewater course. Conclusions: This study indicates that training for all-out runs on a whitewater course is more demanding for canoe slalom athletes than performing all-out trials on a flatwater figure-of-eight course. This evidence may help to develop effective training plans that are essential to reach the highest levels of the sport.

## 1. Introduction

Canoe slalom is an Olympic discipline in which athletes race down a whitewater course in the fastest time possible while having to negotiate a series of gates that are hung over the course. Some gates must be negotiated in a downstream direction and some in an upstream direction, and penalties are accrued for touching or missing the gates. Success in canoe slalom requires the ability to paddle a boat fast and to manoeuvre effectively and precisely through the turbulent water features in order to navigate a route through the gates.

Becoming an elite canoe slalom athlete requires an individual to undertake thousands of hours of training, spread over many years in the sport. The most important factor in performance is technique, especially the paddling technique. Performance tests performed on flatwater are highly correlated with race performance on whitewater [1,2]. However, in canoe slalom, the characteristics of straight paddling in terms of basic biomechanical parameters have still not been described sufficiently. Studies were always conducted on flatwater, focused on the kayaking category only [3,4] or aimed at studying asymmetries [5].

The description of both kinematic (paddling strokes: stroke rate, stroke length; boat acceleration; boat velocity: peak velocity, mean velocity) and kinetic (paddle forces: peak force, mean force, impulse) parameters influencing canoe slalom performance is still lacking, especially on whitewater and in the C1 category (canoe category where athletes kneel in the boat and use single-bladed paddle).

In canoe sprint (flatwater speed canoeing), the biomechanical parameters of paddling have been repeatedly investigated and described (e.g., [6,7,8]), but due to the completely different boats (slalom boats are shorter, slower, turn easier, and have greater hydrodynamic drag), significantly different values can be expected in canoe slalom on the flatwater and even more on the whitewater. Because we do not know the comparison of the forces acting on flatwater and whitewater, we often cannot objectively assess the correct balance between flatwater and whitewater training. Knowing these biomechanical indicators can help to develop an effective training plan which is essential to reach the highest levels of the sport, and this requires an understanding of the correct balance and effectiveness of training on flatwater and on whitewater terrains. We would hypothesise that even when athletes paddle all-out during flatwater or whitewater training, the additional complexity of paddling in whitewater will cause greater biomechanical demands during paddling.

Therefore, the aim of this study was to describe the biomechanical parameters of flatwater and whitewater paddling in men and women in the scientifically less-monitored C1 category.

## 2. Materials and Methods

This study is part of a larger project, with the flatwater methods and initial findings published elsewhere [6]. For this study, twenty C1 canoe slalom athletes were tested (13 males: age 22.0 ± 6.9 years, height 180.0 ± 5.2 m, weight 73.8 ± 8.3 kg; and 7 females: age 18.0 ± 3.4 years, height 168.4 ± 5.9 m, weight 60.4 ± 7.2 kg (mean ± standard deviation)). The athletes were mostly from the Czech junior and senior national teams, with several younger athletes from the local club also participating. Athletes were only considered if they were 15 years or older. All athletes trained regularly and competed internationally in the testing season. The younger athletes were in the Developmental to National level [9] and the older athletes were Elite to World-class [9], including 7 who had won medals at canoe slalom European Championships, World Championships, or Olympic Games. The athletes all provided oral consent to take part in the study, in accordance with requirements from the University Office of Research Ethics. Athlete testing occurred on flatwater and whitewater sections of the canoe slalom training facility at Roudnice na Labem in Czechia.

Athletes initially did their regular warm-up that included dry land stretching and at least 10 min of paddling (technical strokes, short bouts of speed and acceleration, and getting accustomed to the testing equipment). Athletes then paddled two sets of figure-of-eight time trials around two slalom poles that were part of two slalom gates that were hanging above the water; athletes had a 10 min rest between trials. Athlete times were started when their torso passed the first pole, they paddled nine lengths (from one pole to the other), turning to the left around the second pole, and then to the right around the first pole. The time was stopped when their torso passed the second pole at the end of the ninth length. Athletes then paddled 1 or 2 timed runs on the whitewater course (14 downstream and 6 upstream gates), with at least a 15 min rest before each run. The arrangement of gates on a slalom course is changed for every competition and is influenced by the water features at each specific site. For this study, a typical competition course was set by experienced canoe slalom coaches, with the goal of having the fastest time of about 90 s. The performance goal for each of these runs was to achieve the fastest time possible (including 2 s time penalties for touching the gates).

Details of the equipment calibration, data collection, and processing have been described elsewhere [5]. In brief, each athlete paddled their own boat and wore a high-speed satellite positioning system (10 Hz GPS and GLONAS systems, Glo 2, Garmin, Switzerland) strapped to their helmet to measure position and speed, and an inertial measurement unit (recording rate 25 Hz; MetamotionRL, Mbientlab, CA, USA) taped inside their boat to measure boat orientation and acceleration. Paddle forces were measured using strain gauges that were embedded in a spigot that was secured in the paddle shafts between the positions of the top and bottom hands (recording rate 100 Hz; Canoe Power Meter 2nd Gen., One Giant Leap, Nelson, New Zealand), and they were filmed using a 60 Hz video. GPS data were recorded directly onto an Android phone placed in the cockpit, and the IMU and paddle data were stored on these respective devices and downloaded onto a computer after each test. The paddle was equipped with medium-sized blades (Revolution, G’Power, Opatowek, Poland). The strain gauges on the paddle were calibrated at the beginning of the study by hanging weights from the shaft [5] so that the force acting at the centre of the area of the paddle blade could be related to the strain in the paddle shaft and the hand position of the athlete (r^2^ > 0.99).

All data processing was conducted in custom software (Wolfram Research, Inc., Mathematica version 13, Champaign, IL, USA). The baseline strain measurements for the paddles were taken as the mode of the recorded strains, and this is the typical strain during the out-of-water transition phase when the paddle is unloaded. Post-processing divided the data into individual paddle strokes. All types of paddle strokes (on-side or off-side and left or right) were pooled together. Paddle forces were quantified when their absolute force exceeded 60 N. Athletes sometimes used back sweep strokes for the upstream gates, which resulted in negative paddle forces; these strokes accounted for less than 1% of the paddle strokes and were excluded from further analysis. Paddle strokes were quantified by their mean paddle force [N] and impulse [N s] during the pull phase, the mean force [N] during the whole stroke cycle, the pull duration [s] (the time spent pulling on the paddle in the water during the paddle stroke), and the transition duration [s] (the time between the pull phases, most often with the paddle out of the water).

The paddle stroke parameters were visualized by their distributions (subdivided by athletes and flat/white water). For this visualization, the athletes were ranked by the median paddle force that they achieved on the flatwater course. The paddle stroke parameters were statistically evaluated by analysis of variance (ANOVA) using the SPSS version 27 statistical package. Flat/white water was included as a factor, and athlete was included as a random factor. Effects were deemed to be statistically significant at the *p* < 0.05 level. The paddle stroke parameters are described by their estimated marginal means (with standard error of the mean) that emerged from these ANOVAs.

## 3. Results

The mean time to complete the flatwater course was 97.45 ± 7.18 s (mean ± s.d.) and for the whitewater course was 103.41 ± 11.23 s (raw time, not including penalties); the times were not significantly different between the flatwater and whitewater courses. A total of 2106 paddle strokes were measured for the flatwater tests, and 1927 strokes for the whitewater tests (Figure 1 and Figure 2).

The mean force during the pull phase of the paddle strokes was not significantly different between the flatwater and whitewater courses: 138.0 ± 0.57 N and 137.7 ± 0.59 N, respectively. However, there was a significant difference in the duration of the pull phase: 0.48 ± 0.007 s for the flatwater and 0.61 ± 0.007 s for the whitewater course. This led to a significantly lower impulse for the pull phase of 68.8 ± 1.05 N s for the flatwater than 87.2 ± 1.09 for the whitewater course.

The transition duration (between the pull phases) was not significantly different between the flatwater and whitewater courses: 0.45 ± 0.005 s and 0.44 ± 0.005 s, respectively. Thus, the pull phase was a smaller proportion of the whole stroke cycle for the flatwater course, resulting in a lower mean force for the whole stroke cycle of 73.2 ± 0.64 N for the flatwater compared to 81.8 ± 0.66 N for the whitewater course. The paddle stroke frequency (calculated from the combined pull and transition durations) was 1.08 Hz for the flatwater course and 0.95 Hz for the whitewater course.

## 4. Discussion

Comparison of athletes’ metabolic energy sources between straight flatwater and slalom whitewater paddling showed that the energy sources (aerobic versus anaerobic) were remarkably similar, but the absolute metabolic energy costs are difficult to compare due to energy requirement being neither maximal nor constant during whitewater slalom paddling [10]. The mechanical power output during flatwater slalom tests has been determined by using sensors to measure the force, angular velocity, and acceleration in the paddle shaft [3]; however, the mechanical power on whitewater additionally depends on the water velocity, which varies continuously throughout a whitewater slalom course, and so comparisons of mechanical power between flatwater and whitewater courses cannot be made using only such equipment. In this study, we report the mean paddle force and pull phase impulse that are measured from instrumented paddle shafts. These parameters have previously been reported for slalom paddling [3,4,5,11] and can be directly compared between flatwater and whitewater situations.

The female athletes completed run times that were on average 9% slower than the male times. We previously reported [5] that the (flatwater) paddle forces from this study were lower for female athletes and increased with athlete age (that is partly a proxy of experience). Nonetheless, the median paddle forces for each athlete in this report showed a coefficient of determination r^2^ = 0.94 between the flatwater and whitewater scenarios. Thus, athletes with strong paddle forces on flatwater likely also produce strong paddle forces on whitewater, regardless of their sex, age, or experience (Figure 1).

Even though the flatwater course involved both straight sections and turning around gates, the whitewater course was technically more complex due to the addition of the moving water. The athletes would have to continuously negotiate through whitewater features (waves, stoppers, and current differentials) and this requires a more complex set of paddle strokes. Despite this complexity, we found that the mean paddle forces during the pull phase of each stroke were similar for the flatwater and whitewater courses (Figure 1 and Figure 2). The duration of the pull phase was significantly longer for the whitewater course (Figure 1 and Figure 2). In order to accomplish the myriad of different moves on a whitewater course, canoe slalom athletes mostly use a range of turning and blended strokes, which have longer durations than forward strokes [12]. The longer pull phase durations combined with the similar paddle forces resulted in a greater impulse being applied during each pull phase. The transition durations between the pull phases of each stroke were not significantly different between the flatwater and whitewater courses, and thus the longer pull durations occupied a larger proportion of the whole stroke duration for whitewater; this can be termed the duty cycle. Thus, the mean paddle force was greater when expressed over the whole stroke cycle for the whitewater course (Figure 2).

In this study, whitewater testing was always conducted after flatwater testing, leading to a potential bias for fatigue-based reductions in performance on the whitewater courses. Despite this, the mean paddle forces in the drive phase were not reduced, and indeed the overall mean force was higher on the whitewater courses due to the greater duty cycles. Additionally, athletes must endure the additional stress and challenge of the varying water features and currents when on a whitewater course. Successfully negotiating whitewater courses requires additional technical and psychological skills to manage the challenging water conditions; indeed, flatwater performance becomes less of a predictor of whitewater performance as the water difficulty increases [2] and these additional demands become more prominent.

A limitation of this study is that it used one instrumented paddle for all athletes. The length of the paddle was adjusted to be the same as the athlete’s regular paddle; however, the paddle may have had a different blade size, mass, and profile than the athlete’s own paddle (for instance, the instrumentation added 33 g to the mass of the paddle). Thus, the paddle may have been more similar to some athletes’ paddles than to others. This aspect of the paddle performance would contribute to variance in the subject factor used in the ANOVA. However, the main effects of the type of water are independent of this factor and would be largely insensitive to the influence of the paddle. Additionally, all the other parts of the athlete’s equipment were their own, and the testing sessions were kept as realistic as possible.

## 5. Conclusions

During canoe slalom training on all-out trials in a C1 canoe, a greater impulse is used for paddle strokes on whitewater compared to flatwater. This finding suggests that training for all-out runs on a whitewater course is more demanding for canoe slalom athletes than performing all-out trials on a flatwater figure-of-eight course. This provides more support for the suggestion that athletes and coaches should consider the importance of training and preparation races, in advance of important competitions, on whitewater terrain, particularly on water difficulty that resembles where the competitions will be held [2].

## Figures and Tables

**Figure 1 jfmk-09-00167-f001:**
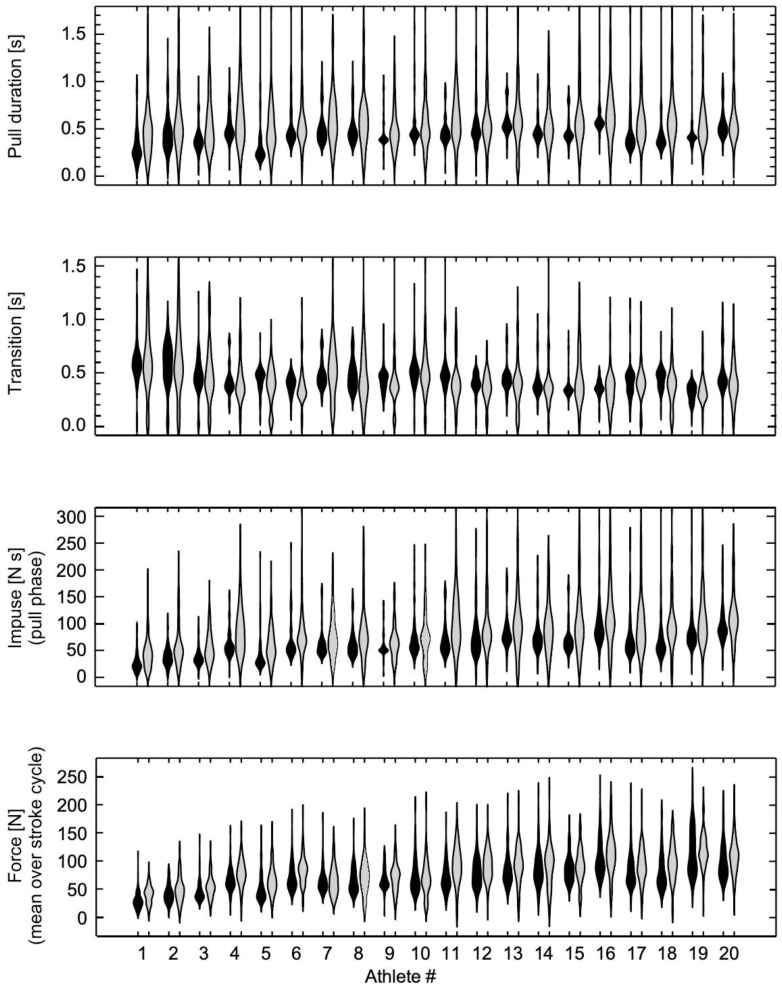
Distribution of pull duration, transition time, impulse, and paddle force for the twenty athletes. Strokes for flatwater training are shown in black, and those for whitewater training are in grey.

**Figure 2 jfmk-09-00167-f002:**
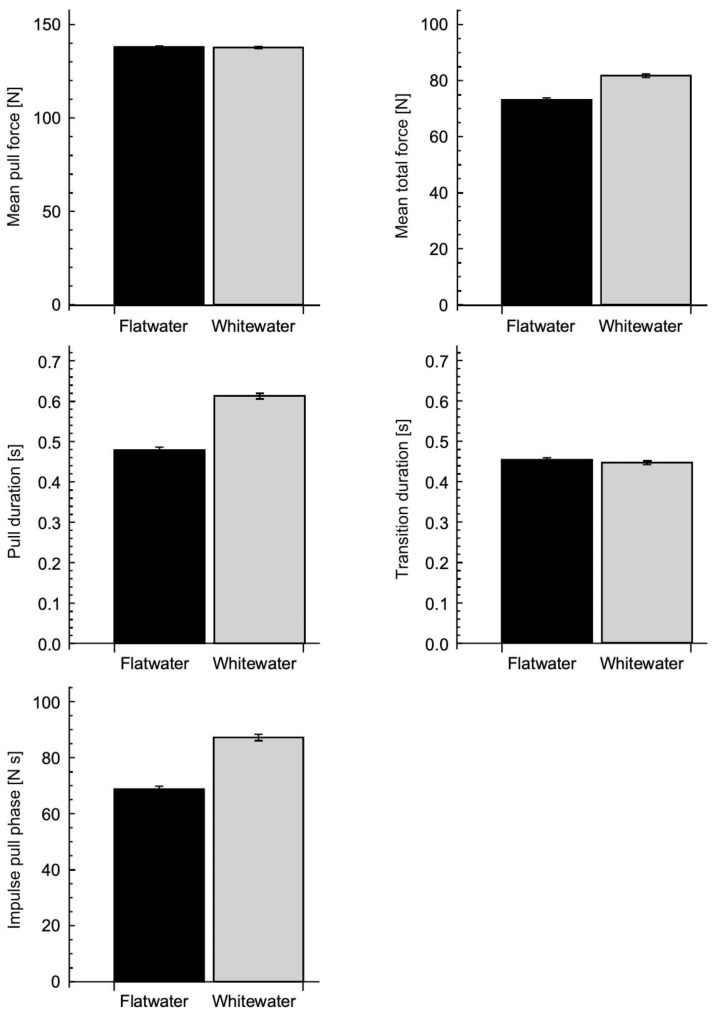
Mean stroke parameters (with standard error of mean). Strokes for flatwater training are shown in black, and those for whitewater training are in grey.

## Data Availability

The original contributions presented in the study are included in the article, further inquiries can be directed to the corresponding author.

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
