# Peer review of "A Comparison of Paddle Forces between Whitewater and Flatwater Training in C1 Canoe Slalom"

_jfmk, 2024, doi:10.3390/jfmk9030167_

Round 1

Reviewer 1 Report

Comments and Suggestions for Authors

The authors aimed to describe paddle forces during canoe slalom training on flatwater and whitewater courses for the C1 canoe category. It is interesting paper for canoe slalom, which offers great value for the slalom coach. The study is well justified. I congratulate the authors for this. 

Perhaps providing some information to facilitate the understanding of the method for canoe foreign readers in relation to courses (the flatwater and whitewater courses) would give added value to the manuscript. It would also answer some doubts that could arise concerning the method.

On one hand, I have some doubts regarding the courses: how were the gates distributed during the whitewater course? What about the order of the downstream and upstream gates? Their placement in the canal? This information can influence your results and it is essential for the reproducibility of your work.

On the other hand, were the duration of the tests (flatwater and whitewater) similar in time?

L145-146: tipo.

I miss some limitations about the study.

Author Response

Thank you for your careful suggestions.

The authors aimed to describe paddle forces during canoe slalom training on flatwater and whitewater courses for the C1 canoe category. It is interesting paper for canoe slalom, which offers great value for the slalom coach. The study is well justified. I congratulate the authors for this. 

Perhaps providing some information to facilitate the understanding of the method for canoe foreign readers in relation to courses (the flatwater and whitewater courses) would give added value to the manuscript. It would also answer some doubts that could arise concerning the method.

An extra description of what the sport involves has been added to the Introduction (lines 30-32)

On one hand, I have some doubts regarding the courses: how were the gates distributed during the whitewater course? What about the order of the downstream and upstream gates? Their placement in the canal? This information can influence your results and it is essential for the reproducibility of your work.

The placement of the gates is described in more detail on lines 86-90. Gate placement is changed for every competition (and for training), and no two whitewater courses are the same. However, we had experienced coaches designing the course in this study, and the test site is one where international competitions are held, and so the course was typical of a canoe slalom competition.

On the other hand, were the duration of the tests (flatwater and whitewater) similar in time?

This information has been added. The tests were not significantly different (lines 130-132). Competition courses are designed so that the fastest kayak times are about 90 seconds. Canoe times are slightly slower. We designed the flatwater course to also meet this target.

L145-146: tipo.

Corrected, thank you.

I miss some limitations about the study.

Limitations have been added (lines 202-211)

Reviewer 2 Report

Comments and Suggestions for Authors

The research makes an important contribution to the field of sports science and biomechanics, particularly within the context of canoe slalom, and it represents a valuable addition to the existing literature. However, several issues require attention before the work is suitable for publication, as detailed below.

The methods section provides a solid framework for understanding the study's execution. Nevertheless, there are a few areas where additional details could improve clarity and comprehensiveness:

- The study mentions the use of strain gauges integrated into the paddle shaft to measure paddle forces. However, it would be beneficial to include more detailed information on the methodology used for these measurements. Specifically, further details about the calibration of the equipment, the sampling rate, and how the data was recorded and analyzed would strengthen the methodological description.

- While the demographic characteristics of the participants are provided, more information on the selection criteria — such as the recruitment process and any inclusion or exclusion criteria — would help contextualize the sample and assess its representativeness.

- The warm-up protocol is mentioned, but additional details regarding the specific exercises and the duration of each component would be useful. Providing this information would be valuable for replicating the study in future research.

The results section clearly presents some key findings, yet there are areas where further detail could enhance the presentation:

-Including additional relevant metrics, such as paddle stroke frequency and variations in technique, would allow for a more comprehensive evaluation of the athletes' performance.

- Exploring individual athlete performance variability within the results would add depth to the analysis. For instance, were there significant differences in performance based on the athletes' experience or gender?

Lastly, while the conclusions are discussed within the text, they are not explicitly presented in a separate section. It is recommended to create a distinct section for the conclusions, where the main results and findings are clearly summarized.

Author Response

Thank you for your careful suggestions.

The research makes an important contribution to the field of sports science and biomechanics, particularly within the context of canoe slalom, and it represents a valuable addition to the existing literature. However, several issues require attention before the work is suitable for publication, as detailed below.

The methods section provides a solid framework for understanding the study's execution. Nevertheless, there are a few areas where additional details could improve clarity and comprehensiveness:

- The study mentions the use of strain gauges integrated into the paddle shaft to measure paddle forces. However, it would be beneficial to include more detailed information on the methodology used for these measurements. Specifically, further details about the calibration of the equipment, the sampling rate, and how the data was recorded and analyzed would strengthen the methodological description.

More details about the data recording and equipment calibration have been added (lines 101-107). The sample rates are also in the methods. The full methods can be seen in reference [7]. All steps of the force analysis are described in lines 108-120: this was conducted in custom-written software.

- While the demographic characteristics of the participants are provided, more information on the selection criteria — such as the recruitment process and any inclusion or exclusion criteria — would help contextualize the sample and assess its representativeness.

More information on the characteristics of the athletes have been added (lines 68-70).

- The warm-up protocol is mentioned, but additional details regarding the specific exercises and the duration of each component would be useful. Providing this information would be valuable for replicating the study in future research.

The warm up followed each athletes normal routine so was specific to each athlete. The one criterion was that they warmed up for at least 10 minutes. The types of warm up has similar aspects between athletes, and more information describing this has been included (lines 77-79).

The results section clearly presents some key findings, yet there are areas where further detail could enhance the presentation:

-Including additional relevant metrics, such as paddle stroke frequency and variations in technique, would allow for a more comprehensive evaluation of the athletes' performance.

Paddle stroke frequency for the flatwater and whitewater courses are now reported (line 152-153). General differences in technique can be seen in the distribution of the metrics in Figure 1. However, we cannot identify the different types of paddle stroke from the strain data on the paddle shafts, and have had to pool the different types of paddle stroke (line 112-113) and so cannot comment further on possible differences in technique.

- Exploring individual athlete performance variability within the results would add depth to the analysis. For instance, were there significant differences in performance based on the athletes' experience or gender?

Experience and gender differences are now described. However, the main focus remains the comparison between flatwater and whitewater metrics, regardless of the experience and gender of the athlete (lines 169-175).

Lastly, while the conclusions are discussed within the text, they are not explicitly presented in a separate section. It is recommended to create a distinct section for the conclusions, where the main results and findings are clearly summarized.

An explicit Conclusion section has now been added (lines 212-220).

Round 2

Reviewer 2 Report

Comments and Suggestions for Authors

I'd like the Authors to have properly addressed all Reviewer's comments